# Rational Sampling Numbers of Soil pH for Spatial Variation: A Case Study from Yellow River Delta in China

Yingxin Zhang [1,†], Mengqi Duan [1,†], Shimei Li [2,*], Xiaoguang Zhang [1,3,*], Xiangyun Song [1] and Dejie Cui [1]

1   Department of Resources and Environment, Qingdao Agricultural University, Qingdao 266109, China; zyx1031644968@163.com (Y.Z.); duanmengqi28@163.com (M.D.); songxiangyun2000@126.com (X.S.); cuidejie@163.com (D.C.)
2   Department of Landscape Architecture and Forestry, Qingdao Agricultural University, Qingdao 266109, China
3   Qingdao Engineering Research Center for Remote Sensing Application in Agriculture, Qingdao 266109, China
*   Correspondence: li_shimei@163.com (S.L.); zhangxg_66@sina.com (X.Z.); Tel.: +86-158-6309-1676 (S.L.); +86-151-9200-3056 (X.Z.)
†   These authors contributed equally to this work.

**Abstract:** Spatial variation of soil pH is important for the evaluation of environmental quality. A reasonable number of sampling points has an important meaning for accurate quantitative expression on spatial distribution of soil pH and resource savings. Based on the grid distribution point method, 908, 797, 700, 594, 499, 398, 299, 200, 149, 100, 75 and 50 sampling points, which were randomly selected from 908 sampling points, constituted 12 sample sets. Semi-variance structure analysis was carried out for different point sets, and ordinary Kriging was used for spatial prediction and accuracy verification, and the influence of different sampling points on spatial variation of soil pH was discussed. The results show that the pH value in Kenli County (China) was generally between 7.8 and 8.1, and the soil was alkaline. Semi-variance models fitted by different point sets could reflect the spatial structure characteristics of soil pH with accuracy. With a decrease in the number of sampling points, the Sill value of sample set increased, and the spatial autocorrelation gradually weakened. Considering the prediction accuracy, spatial distribution and investigation cost, a number of sampling points greater than or equal to 150 could satisfy the spatial variation expression of soil pH at the county level in the Yellow River Delta. This is equivalent to taking at least 107 sampling points per 1000 km$^2$. The results in this study are applicable to areas with similar environmental and soil conditions as the Yellow River Delta, and have reference significance for these areas.

**Keywords:** sampling; soil pH; spatial variation; ordinary kriging

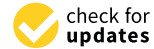



## 1. Introduction

Soil pH is an important index for evaluation of land quality [1]. Soils with acid and alkali over the national standard are not conducive to the utilization of land resources [2]. Therefore, it is very important to understand the spatial distribution of soil pH. At the same time, it is of great guiding significance to accurately grasp the spatial variation characteristics of soil pH for evaluating the salinization and acidification of the soil environment, rational fertilization and efficient utilization of nutrients [3]. However, sampling number affects the accuracy of soil properties and their spatial variation information and the degree of quantitative expression. The layout pattern and sampling number must be fully considered to ensure the accuracy of spatial interpolation in any study on spatial variation of soil pH [4–6]. Generally speaking, the larger the sampling density, the smaller the sample error and the higher the accuracy of the research results; however, this means the work cycle will be prolonged, and huge manpower, material resources and financial resources will be consumed [7]. Sampling costs also limit the sampling density to a large extent. If the number of sampling points is reduced, the interpolation accuracy of soil pH

spatial variation will be difficult to guarantee, and local features may not be displayed [8,9]. Therefore, it is of great significance to study the reasonable number of sampling points for accurate quantitative expression on spatial distribution of soil pH and resource savings.

Using spatial geostatistics is one of the most accurate spatial prediction methods, and is often used to model the spatial variability for soil properties and evaluate their spatial uncertainty [10]. At present, a method combining geostatistics and GIS is used to study the spatial distribution of regional soil properties from the perspective of spatial prediction, and is also used to analyze the influence of sampling density on the spatial variation of soil properties [11]. In recent years, many scholars have carried out much of geostatistical research on the influences of sampling number on spatial variability for different soil properties in different areas, such as soil organic matter [12–14], nitrogen [15], exchangeable potassium, calcium, magnesium [16], heavy metals [17,18] and salt. [19]. The reasonable sampling quantity of different soil properties is basically different for their different characteristics. Even for the same specific soil index, the results of different research are different. For example, for soil organic matter, the reasonable sampling points of spatial variation in typical areas of Yangtze River Delta were 91 per 1000 km$^2$ [20], and 547 per 1000 km$^2$ in Fei County, which is a typical county of North China Plain [21]. For soil organic carbon, 908 sampling points were reasonable in typical gully areas of the Loess Plateau, which means that 178 sampling points were needed per 1000 km$^2$. These studies show that there were differences in the number of reasonable sampling points even with the same soil index because of the differences in the natural geographical environment, such as topography and geomorphology in different areas.

In addition, survey scale has a certain influence on reasonable sampling number. For example, in relatively consistent geomorphic units, such as Fujian Province and its counties, the reasonable sampling points of spatial distribution for soil organic matter were 10,000 and 11,000 for every 1000 km$^2$, respectively [12]. The existing research methods and conclusions still need to be tested because of the different evaluation indexes, the different natural geographical conditions and the different influence of human activities in different study areas.

To sum up, although many scholars have carried out relevant research on reasonable sampling numbers, at present there are few reports on related research for soil pH. Especially, research in county-level areas is lacking on the influence of different sampling points on the spatial variation of soil pH, and because of the fragile and salinized soil environment in the Yellow River Delta region and great attention from the government, it is of great significance to monitor and master the spatial distribution status of soil pH for green development in agriculture on the premise of clarifying the reasonable sampling numbers. Therefore, Kenli County in the Yellow River Delta was selected as the study area, and the spatial distribution of soil pH with 12 different sampling sets was predicted using geostatistical Kriging interpolation. The overall objectives of this research were (1) to assess the influence of different sampling numbers on the prediction accuracy of spatial distribution for soil pH, and (2) to determine reasonable sampling densities to determine the spatial variation of soil pH at the county scale.

## 2. Materials and Methods

### 2.1. Study Area

Kenli County in the Yellow River Delta region was selected as the study area (Figure 1). This county is located at the mouth of the Yellow River in the Yellow River Delta of northern Shandong Province in China, between 37°24′–38°10′ N and 118°15′–119°19′ E [22], with a total area of 2331 km$^2$. It has a temperate monsoon climate, with high annual temperature and uneven spatial-temporal distribution of precipitation. The terrain is fan-shaped and slightly inclined from southwest to northeast [23]. The altitude ranges from 2 m to 11.61 m. The parent material is loess. The mechanical composition of the soil is mainly sandy loam. The main soil types in Kenli County are fluvo-aquic soil and coastal saline soil in the soil genetic classification of China. The corresponding soil group names

from WRB are Cambisols and Solonchaks, respectively. The typical crops are cotton, rice and wheat-corn rotation. Kenli County has rich soil resources, and is one of the most abundant land reserve resources in the coastal areas of eastern China, with great potential for agricultural development.

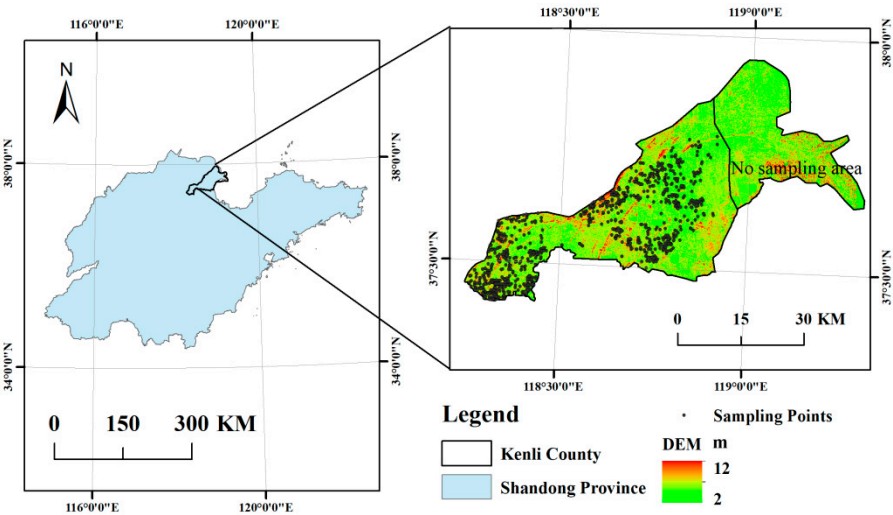

**Figure 1.** Overview of the study area.

## 2.2. Collection and Processing of Sample Data

Based on the grid distribution point method, 1140 sampling points were used in this research. Some points on non-agricultural land were deleted and 1000 points were left. These sampling points were mainly distributed in cultivated land, and the interval between sample points was about 1000 m. In the actual sampling process, we adjusted the specific positions of these sampling points for road accessibility and crop planting. The topsoil from 0–20 cm was taken with a shovel, and then was put into a plastic bag and brought back to the laboratory for analysis. Although 926 samples were sampled, there were many uncertainties and complexity in field sampling. Consequently, we eliminated some points that were not standardized in the collection process, and removed some outliers. Finally, 908 samples were obtained, and 797, 700, 594, 499, 398, 299, 200, 149, 100, 75 and 50 sampling points were randomly selected from these 908 sampling points (Figure 2). The above real numbers represent the approximate number sets of 900, 800, 700, 600, 500, 400, 300, 200, 150, 100, 75, 50, respectively. Each sampling point was extracted from the last sampling point set to compare the characteristics of different sample sets. For example, the 700 sample set was extracted from the 797 sample set, and the 299 sample set was extracted from the 398 sample set. Combining all the sample data, a total of 12 sample sets were formed. The sampling process was conducted by the "Geostatistical Analyst" module of arcgis 10.0. The pH of each sample was measured in 1:2.5 mixtures of soil and deionized water with a pH meter by a potentiometric method [24].

## 2.3. Spatial Prediction and Verification Method

Geostatistical methods are widely used to predict the spatial distribution of soil properties [25–29]. In this paper, we chose the Ordinary Kriging (OK) method to predict the spatial distribution of soil pH [30]. The OK method satisfies the intrinsic hypothesis, and the average value of the regionalized variables is an unknown constant [31]. OK is a linear estimation of regionalized variables, which is similar to weighted moving average in the process of interpolation research. However, the weights of weighted moving average are determined from different sources. The weighted sliding average weight values are derived from known spatial functions, while the weights of ordinary kriging are derived from spatial data analysis [31]. It is necessary to verify the prediction results after spatial distribution prediction. In this paper, an independent verification method was adopted.

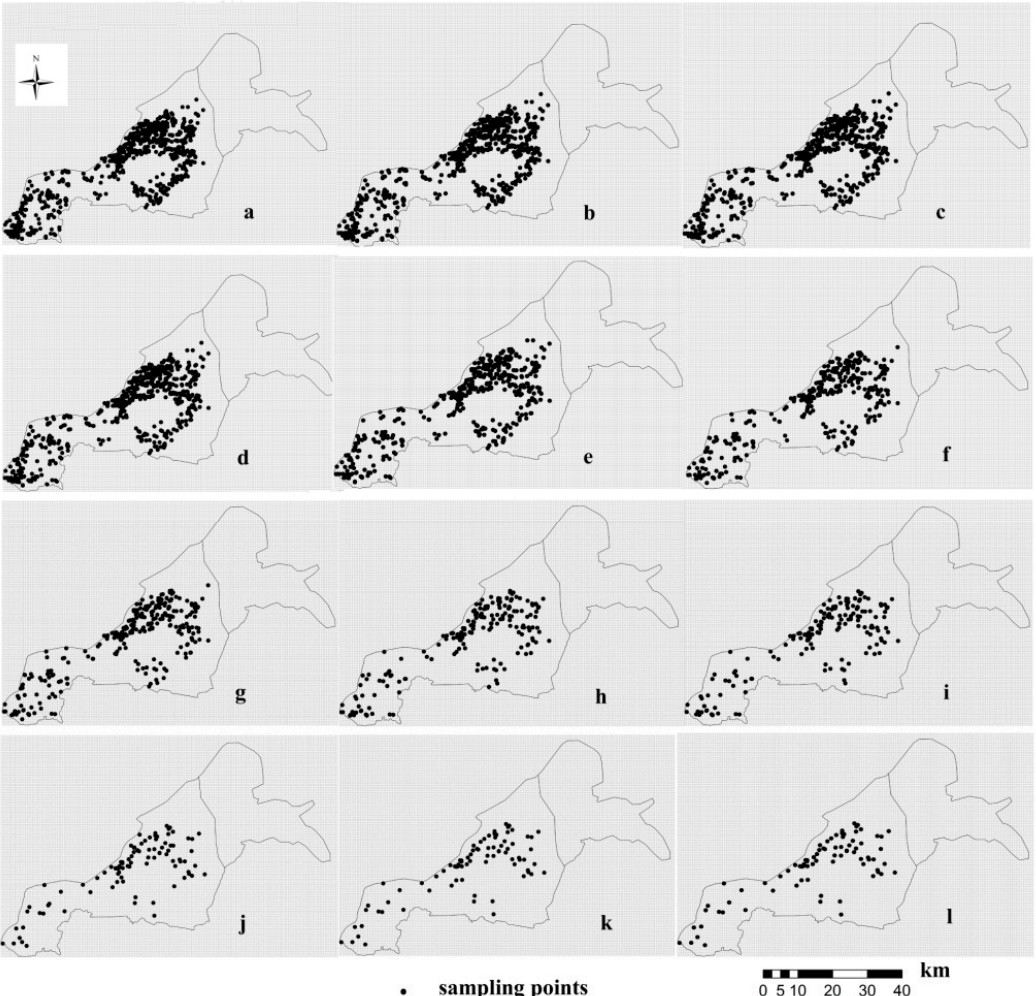

**Figure 2.** Distribution map of soil pH at different point sets. The letters (**a**–**l**) represent the spatial variation expression of soil pH at 908, 797, 700, 594, 499, 398, 299, 200, 149, 100, 75 and 50 sampling points.

The OK interpolation results were verified. The verification method of independent data set extracts some samples from all samples as independent data sets, takes the remaining samples as simulation data sets without repetition, and takes each sample in independent data sets as an inspection point [32]. In this paper, after 797 samples were extracted from 908 samples, the remaining 111 samples were taken as independent verification sets. The spatial prediction results of 12 sample sets were verified. In the verification method of the independent data set, the most representative evaluation indexes are root mean square error (RMSE), mean error (ME) and average standard error (ASE), which were chosen to evaluate the accuracy of prediction.

$$\text{RMSE} = \sqrt{\frac{1}{N} \sum_{i=1}^{n} \left[ Z(X_i) - Z'(X_i) \right]^2} \tag{1}$$

$$\text{ME} = \frac{1}{N} \sum_{i=1}^{n} \left[ Z(X_i) - Z'(X_i) \right] \tag{2}$$

$$\text{ASE} = \sqrt{\frac{1}{N} \sum_{i=1}^{n} \left[ Z'(X_i) - \sum_{i=1}^{n} (Z'(X_i))/N \right]^2} \tag{3}$$

where $N$ is the number of known samples, the actual value is $Z(X_i)$, and the estimated value is $Z'(X_i)$.

The smaller the RMSE, the closer the ME to zero, indicating that the accuracy of spatial prediction is higher. The average standard error was used to measure the uncertainty of the Kriging prediction value.

## 3. Results

### 3.1. Descriptive Statistics Characteristics of Soil pH for Different Sets of Sample Points

Descriptive statistics analysis was made on 12 sample sets, and the results are shown in Table 1. The soil pH values of 908 sampling points in the study area ranged from 7.00 to 8.80, with an average value of 7.85. The soil was weakly alkaline. The coefficient of variation was 5.35%, and the variability was weak, indicating that the alkalization degree was very concentrated. The skewness coefficient was −0.31, and the kurtosis coefficient was 2.12. The results of normal test on skewness coefficient and kurtosis coefficient showed that the soil pH values of different sampling points were in accordance with normal distribution.

**Table 1.** Descriptive statistics characteristics of soil pH for different sets of sample points.

| Sampling Point Number | Min (g/kg) | Max (g/kg) | Average (g/kg) | Standard Deviation (g/kg) | Skewness | Kurtosis | Median (g/kg) | Variation (%) |
|---|---|---|---|---|---|---|---|---|
| 908 | 7.00 | 8.80 | 7.85 | 0.42 | −0.31 | 2.12 | 7.90 | 5.35 |
| 797 | 7.00 | 8.80 | 7.84 | 0.42 | −0.32 | 2.15 | 7.90 | 5.36 |
| 700 | 7.00 | 8.80 | 7.85 | 0.41 | −0.33 | 2.18 | 7.90 | 5. 22 |
| 594 | 7.00 | 8.80 | 7.85 | 0.42 | −0.35 | 2.18 | 7.90 | 5.35 |
| 499 | 7.00 | 8.80 | 7.86 | 0.41 | −0.37 | 2.19 | 7.90 | 5.22 |
| 398 | 7.00 | 8.80 | 7.88 | 0.41 | −0.37 | 2.18 | 7.90 | 5.20 |
| 299 | 7.00 | 8.80 | 7.88 | 0.40 | −0.38 | 2.30 | 7.90 | 5.08 |
| 200 | 7.00 | 8.80 | 7.89 | 0.42 | −0.43 | 2.33 | 7.90 | 5.32 |
| 149 | 7.00 | 8.60 | 7.90 | 0.42 | −0.59 | 2.46 | 8.00 | 5.32 |
| 100 | 7.10 | 8.80 | 7.90 | 0.40 | −0.35 | 2.40 | 7.95 | 5.06 |
| 75 | 7.00 | 8.60 | 7.88 | 0.40 | −0.55 | 2.43 | 8.00 | 5.08 |
| 50 | 7.10 | 8.60 | 7.85 | 0.41 | −0.15 | 2.07 | 7.90 | 5.22 |

The minimum value of soil pH for 100 and 50 sampling points was 7.10 g/kg, and the minimum value of other sampling points was 7.00 g/kg. The maximum values of 908, 797, 700, 594, 499, 398, 299, 200 and 100 sample points were all 8.80 g/kg, and the maximum values of only 149, 75 and 50 sample points were 8.60 g/kg. However, they were still very similar. Among the 11 sub-samples, the average value and standard deviation of soil pH also fluctuated around the average value and standard deviation of the complete set, which indicated that although the number of sampling points decreased, the 11 samples could still represent the complete set. The coefficient of variation of soil pH ranged from 5.06% to 5.36% among the 11 sub-samples, and the variability was weak. To sum up, the analysis of each index of each subset showed that the selected subsets were all representative.

### 3.2. The Influence of Different Sampling Points on the Semi-Variance Structure of Soil pH

Semi-variance analysis of soil pH was carried out by a geostatistical method. Table 2 shows the semi-variance function values of soil pH under different sampling points. The spatial variation structure of soil pH at other sampling densities conformed to the exponential model, except for 75 sampling points. The level of decision coefficient represented the effect of fitting the variogram by the model. The higher the decision coefficient, the better the effect of fitting the variogram by the model [33]. The determination coefficients of different sampling points were between 0.39 and 0.69, indicating that the model could reflect the spatial structure characteristics of soil pH with accuracy.

The ratio of Nugget to base Sill (C0 + C) reflects the degree of spatial autocorrelation of variables. This is considered a strong spatial autocorrelation when the ratio is less than 25%, has moderate spatial autocorrelation when the ratio is between 25% and 75%, and has weak spatial autocorrelation when the ratio is greater than 75%. The Nugget/Sill of

the total sample set (908 sample points) in this study was less than 25%, showing a strong spatial autocorrelation. The Nugget/Sill of 200–800 samples in the other 11 sample subsets was less than 25%, which indicates that these sample sets had strong spatial autocorrelation. However, when the number of samples was less than 200, the Nugget/Sill ranged from 25% to 75% (with moderate spatial autocorrelation). When the number of samples was less than 150, the Nugget/Sill reached 40%, which indicates that the spatial autocorrelation of these samples was weakened.

**Table 2.** Parameters of semi-variance function of soil pH under different sampling points.

| Sampling Point Number | Model | Nugget (C0) | Sill (C0 + C) | C0/Sill (%) | Range (km) | Determination | Residual |
|---|---|---|---|---|---|---|---|
| 926 | exponential model | 0.0350 | 0.1786 | 19.60 | 4.80 | 0.68 | 0.00 |
| 801 | exponential model | 0.0360 | 0.1778 | 20.25 | 5.13 | 0.72 | 0.00 |
| 704 | exponential model | 0.0360 | 0.1752 | 20.55 | 4.77 | 0.68 | 0.00 |
| 598 | exponential model | 0.0360 | 0.1780 | 20.22 | 4.75 | 0.69 | 0.00 |
| 502 | exponential model | 0.0330 | 0.1742 | 18.94 | 4.82 | 0.69 | 0.00 |
| 401 | exponential model | 0.0350 | 0.1750 | 20.00 | 4.70 | 0.60 | 0.00 |
| 300 | exponential model | 0.0350 | 0.1658 | 21.11 | 4.81 | 0.60 | 0.00 |
| 201 | exponential model | 0.0295 | 0.1810 | 16.30 | 4.87 | 0.44 | 0.00 |
| 150 | exponential model | 0.0442 | 0.1736 | 25.44 | 6.54 | 0.39 | 0.00 |
| 100 | exponential model | 0.0952 | 0.1914 | 49.74 | 26.52 | 0.64 | 0.00 |
| 75 | spherical model | 0.0869 | 0.1748 | 49.71 | 13.77 | 0.49 | 0.00 |
| 50 | exponential model | 0.1247 | 0.3174 | 39.29 | 160.59 | 0.61 | 0.00 |

The variable range represents the autocorrelation range of the variogram [34], and can reflect the size of the autocorrelation range in the variable space. In this paper, the fitting range of soil pH under a different number of points was more than 4 km, indicating that the spatial autocorrelation distance was relatively large. Among them, when the number of sampling points was reduced to 100, the range increased, reaching 26.52, which was about five times that of the 200 sampling points.

### 3.3. The Influence of Different Sampling Points on the Spatial Prediction Accuracy of Soil pH

For each sample set, Kriging interpolation was used to carry out the spatial prediction of soil pH. Root mean square error (RMSE), mean error (ME) and average standard error (ASE) were used to measure the prediction accuracy of soil pH under different sampling points.

It can be seen from Figure 3 that ME of 75, 100, 150, 200 and 300 was greater than 0, and that of other sampling points was less than 0. The ME value varies with the number of sampling points, but the variation does not follow the law that the ME decreases with the increase of sample number. When the sampling points were 50 and 300, the ME values were close to 0. In theory, the closer the ME to 0, the higher the accuracy of spatial prediction. In this case, the value of ME is generally calculated from statistical methods. It is possible that the independent verification results of each sample point were poor, but the residual errors after addition and averaging were smaller. When ME is close to 0, the prediction has low accuracy. Therefore, the ME index cannot indicate the accuracy very well at this point. On the other hand, a single index cannot indicate the interpolation accuracy well, and multiple indexes may be more accurate to judge the interpolation accuracy. With the decreased of sample size, the distance of ME deviating from X axis first increased, then decreased and then slightly increased. It basically surrounded the X axis except for 75 and 100 sample points. Therefore, when the number of sample points was too small (75, 100), the ME of predicted values would become larger.

The RMSE was slightly larger with 50 samples. There was no obvious change trend with the decreased sample size, and it could remained at a certain value. This shows that the RMSE had no obvious difference in the spatial prediction accuracy of different

sample subsets, i.e., the accuracy of Kriging interpolation had no significant difference with decreased sample size.

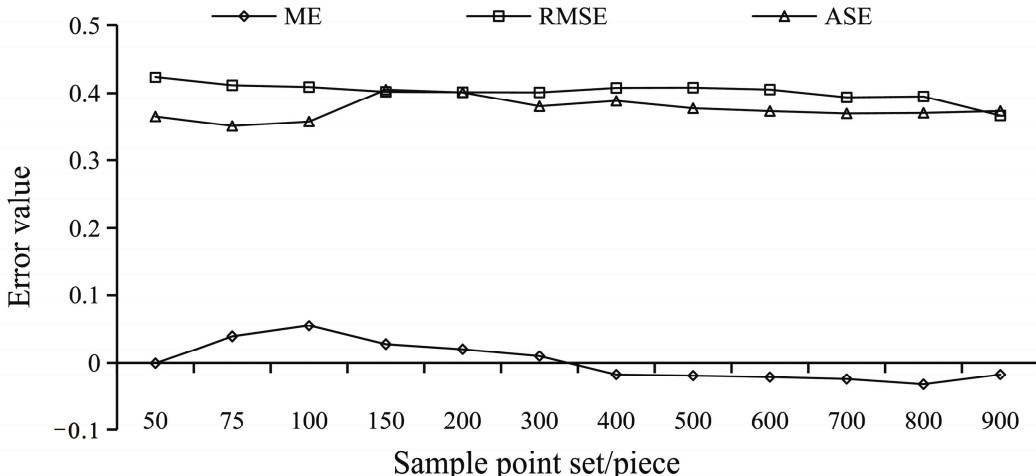

**Figure 3.** Prediction error of soil pH at different point sets.

The uncertainty of Kriging prediction was measured by the ASE. The closer the ASE the RMSE, the more accurate the prediction of attribute value. In this study, when there were 150, 200 and 900 samples, ASE and RMSE were basically equal, indicating that the spatial variability of the earth was properly estimated. When the number of sampling points was 300–800, the ASE was smaller than the RMSE, and the difference between them was very small, indicating that these sampling points overestimated the spatial variability. However, when the number of sampling points was less than 150, the ASE was obviously less than the RMSE, so these sample sets could not reasonably predict the variability.

Considering the ME, RMSE and ASE, 107 sampling points per 1000 km$^2$ could meet the needs of spatial variation expression of soil pH in the Yellow River Delta. According to the above analysis, we also determined that it was not enough to evaluate Kriging prediction accuracy only by a single evaluation index. Therefore, it was necessary to use a variety of evaluation indexes and combine them to accurately evaluate the prediction results.

### 3.4. Effects of Different Sampling Points on Spatial Distribution of Soil pH

To more intuitively show the influence of different sampling point sets on the spatial distribution of soil pH, the Kriging method was used to carry out a spatial interpolation operation (Figure 4). The eastern of Kenli County was not sampled because the area is in the Yellow River Delta National Nature Reserve. However, for the continuity and completeness of pictures, Kriging interpolation was extended to the entire Kenli County. When analyzing the spatial variation characteristics of soil pH, the Yellow River Delta National Nature Reserve in the east of Kenli County was not a focus of analysis.

In the study area, the yellow part shown on the map (Figure 4) had a large area: that is, the soil pH in Kenli County of the Yellow River Delta was generally between 7.8 and 8.1, which proved that the soil in this area was alkaline. The areas with high soil pH value (orange-red) were distributed in the middle and southwest of study area. With the decreased of the number of sample points, the ability to describe details was gradually weakened. When the number of sampling points was reduced to 100, the details of soil pH in the middle and north of the study area were no longer detailed, and only a general distribution trend could be seen. Therefore, considering the precision and research funds, it is suggested that the reasonable sampling number in the Yellow River Delta should not be less than 150.

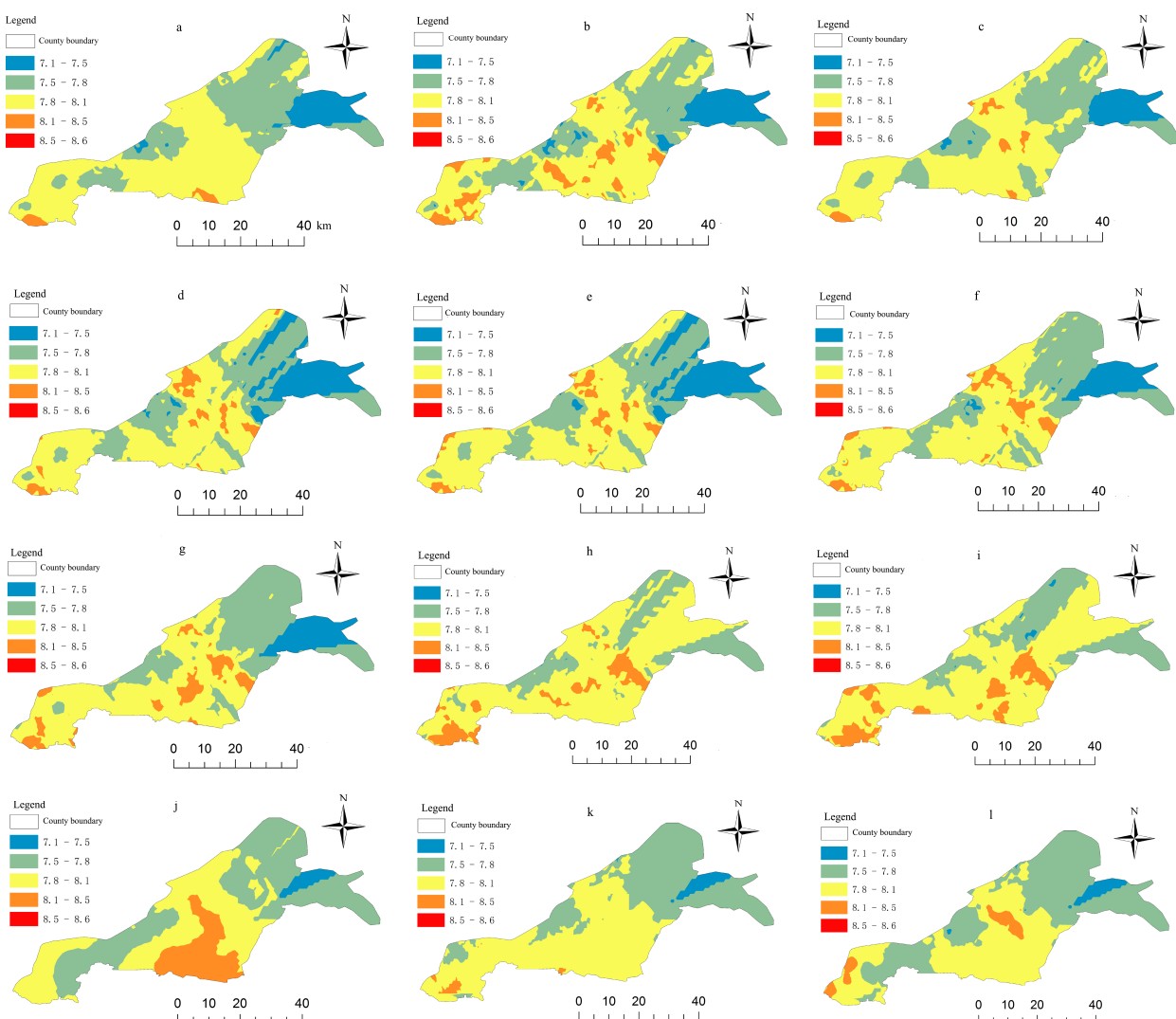

**Figure 4.** Expression diagram of soil pH spatial variation of different point sets. The letters (**a**–**l**) represent the spatial variation expression of soil pH at 908, 797, 700, 594, 499, 398, 299, 200, 149, 100, 75 and 50 sampling points respectively. Note: The eastern of Kenli County was not sampled because the area is in the Yellow River Delta National Nature Reserve. However, for the continuity and completeness of maps, the result of the unsampled area was deduced and extended by the Kriging interpolation based on the near sampled soils.

## 4. Discussion

Spatial variation of soil properties is controlled by various structural and random factors. The larger the ratio of Nugget to Sill, the more obvious the influence of human activities, such as irrigation, fertilization and cultivation. On the contrary, structural factors such as soil parent material, climate, biology, topography and other natural factors play a major role [35,36]. The ratio of Nugget to Sill of almost all sample sets in this study was less than 25% and showed strong spatial autocorrelation, indicating that the spatial variation of soil pH in the study area was mainly affected by structural components such as topography, climate and soil parent material. When the sampling point was less than 150, the ratio of Nugget to Sill reached 40%, indicating that the spatial autocorrelation of these sample sets was weakened, and was influenced by structural components and random factors. With the decreased of the number of sampling points, the small-scale structural factors and random factors gradually increased, while the influence of large-scale structural factors such as parent material, topography and soil type on soil pH gradually weakened, which caused the soil pH to change strongly with the small number of sampling points.

According to the analysis of spatial prediction accuracy of soil pH with different sampling sets, the ME of predicted values gradually increased when the number of sampling points was too small. The RMSE could basically be maintained at a certain value with a decrease of sampling numbers, except for being slightly larger at 50 sample points. Interpolation prediction error and interpolation of the distribution map had a certain synergistic relationship with the change of sample numbers. When there were less than 150 samples, the difference between ASE and RMSE was large. Correspondingly, the spatial distribution maps of soil pH with less than 150 sampling points were too smooth to show the spatial distribution of soil pH accurately. However, when the number of samples was increased to 150 or more, details of the spatial variation expression diagram of soil pH were more obvious, and could show the spatial variation expression of soil pH more accurately. In addition, at 150 or more sampling points, the prediction error maps of RMSE, ME and ASE of soil pH and the expression maps of spatial variation of soil pH in different point sets could better predict the spatial variation of soil pH. Therefore, it was shown that the prediction error of soil pH was consistent with the spatial distribution of soil pH.

In this paper, the influence of the number of different sampling points on the spatial distribution of soil pH in the Yellow River Delta region was investigated by using the ordinary Kriging method. It was concluded that the number of sampling points most suitable for Kenli County in the Yellow River Delta region should be no less than 150; that is, at least 107 sampling points should be taken every 1000 km$^2$. This paper also compared the existing literature investigating reasonable sampling numbers of soil pH. Study [37] showed that the rational sampling number at the county scale was about 4900 samples per 1000 km$^2$, which was inconsistent with our findings. This may be because in that study the area was located in a more undulating mountainous and hilly region resulting in a reduced spatial autocorrelation of soil pH. While this result was similar to the most reasonable number of sampling points needed for soil salinity in Kenli County of the Yellow River Delta studied by Zhang et al. [19], it revealed that the number of sampling points for spatial variation expression of different soil properties in areas with similar environmental conditions may get closer.

## 5. Conclusions

In this paper, Kenli County in the Yellow River Delta was selected as the research area. Twelve sample sets consisting of 908, 797, 700, 594, 499, 398, 299, 200, 149, 100, 75 and 50 sampling points were selected to study the influence of sampling number on the spatial variation of soil pH. A reasonable sampling number was determined, which provides for the collection of soil samples with minimum human, material and financial resources for research on soil pH.

With the decreasing the sampling points number, the C0/Sill value of the sample set increased, and the spatial autocorrelation decreased gradually. The variation range of soil pH fitted by different numbers of points was greater than 4 km, and the spatial autocorrelation distance was relatively large. When the number of sampling points was reduced to 100, the range increased significantly, reaching 26.52, which was about 5 times that of 200 sampling points. Comprehensive analysis of ME, RMSE and ASE showed that when the number of sampling points was 150, prediction accuracy was the highest, which can satisfy the spatial variation expression of soil pH in the Yellow River Delta region.

The pH value in Kenli County was generally between 7.8 and 8.1, and the soil was alkaline. Areas with high soil pH value were distributed in the middle and southwest of the study area. With a decreased number of sampling points, the detailed characteristics of spatial variation of soil pH gradually disappear. When the number of sampling points was 150, it could not only describe the spatial distribution of soil pH in detail, but also accurately describe the spatial distribution pattern of soil pH.

Therefore, considering prediction accuracy, spatial distribution and research funding, it is suggested that the reasonable sampling number should be no less than 150 in the Yellow River Delta region, which is equivalent to at least 107 sampling points for 1000 km$^2$.

The results of this study are applicable to areas with environmental and soil conditions similar to those in the Yellow River Delta and have reference significance for these areas. However, the number of sampling points will be different in other areas to reasonably express the spatial distribution of soil pH, and needs to be analyzed in combination with local environmental conditions.

**Author Contributions:** Conceptualization, Y.Z. and X.Z.; data curation, Y.Z. and M.D.; investigation, S.L. and X.Z.; methodology, M.D.; project administration, X.Z.; software, M.D.; supervision, S.L., X.Z. and X.S.; validation, M.D.; visualization, Y.Z.; writing–original draft, Y.Z.; writing–review & editing, X.Z., X.S. and D.C.; funding acquisition, X.S. and D.C. All authors have read and agreed to the published version of the manuscript.

**Funding:** This research was funded by the National Key Research and Development Program of China (grant number 2021YFD1900900), the Key Research and Development Program of Shandong Province, China (grant number 2021CXGC010801, 2021CXGC010804), Shandong Province Modern Agricultural Industry Technology System Cotton Post Innovation Team (grant number SDAIT-03-06), and the Talent Fund of Qingdao Agricultural University, China (grant number 1114344).

**Conflicts of Interest:** The authors declare no conflict of interest.

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
