# Peer review of "Rational Sampling Numbers of Soil pH for Spatial Variation: A Case Study from Yellow River Delta in China"

_applsci, doi:10.3390/app12136376_

Round 1
Author Response
Responses to the comments of Reviewer #1
Dear Reviewer,
Thank you very much for your important comments, which will contribute much to the improvement of the manuscript.
In the revised version, we have made revisions in accordance with the points raised by you. We hope these changes would meet the requirements.
Yours sincerely,
Xiaoguang Zhang
Our replies to your comments are here as below:
Title
Comment 1: I suggest authors to mention in the title the country where this study was done Rational sampling numbers of soil pH for spatial variation: A case study from Yellow River Delta in China.
Response 1: Thank you for your careful suggestion. According to your suggestions, we have changed the title to "Rational sampling numbers of soil pH for spatial variation: A case study from Yellow River Delta in China".
Abstract
Comment 1: Precise the country in the abstract text
... The results showed that the pH value in Kenli County (China) ...
Response 1: Thank you for your suggestion. As your suggestion, we have modified the abstract to make the country precise (Line 21-22, Page 1) as follow:
... The results showed that the pH value in Kenli County (China) was generally between 7.8 and 8.1 ...
Introduction
Comment 1: Line 59: ter[10,11,12], nitrogen [13], exchangeable potassium, calcium and magnesium[14] and Typing error, put a space before bracket like ... nitrogen [13]. The same remark for the whole text Line 77 hman activities in different study areas. Typing error in hman, it misses "u" (human)
Line 88-89: ...1) to clarify the influence of different sampling numbers on 88 the prediction accuracy of spatial distribution for soil pH;
I suggest to author to use "to assess" instead of clarify
Response 1: Thank you for your valuable suggestion. According to your suggestions, we have put a space before bracket (Line 68-69 and 75, Page 2) and corrected typing error to "human" (Line 87, Page 2). And we have used "to assess" instead of "clarify" (Line 98, Page 2).
Materials and methods
Comment 1: Line 94-95 ... indicate the country in this sentence as follow ... This county is located at the mouth of the Yellow River in the Yellow River Delta of northern Shandong Province in China
Response 1: According to your suggestions, we have indicated the country in this sentence as follow (Line 113, Page 3):
... This county is located at the mouth of the Yellow River in the Yellow River Delta of northern Shandong Province in China ...
Comment 2: Line 105-106
Based on the grid distribution point method, more than 1 100 sampling points were distributed in this study.
Does the distribution of points follow a scale? It should be better to provide more information about the distribution of points, even provide a map of points distribution
Response 2:
According to your suggestions, we have provided more information about the distribution of points in this sentence as follow (Line 123-129, Page 3):
Based on the grid distribution point method, 1 140 sampling points were set in this research. These sampling points were mainly distributed in cultivated land, and the interval between sample points was about 600 m. In the actual sampling process, we have adjusted the specific positions of these sampling points for road accessibility and crop planting. Although all these samples were sampled, there were many uncertainties and complexity in field sampling. Consequently, we eliminated some points that were not standardized in the collection process, and removed some outliers, obtained 908 sampling points. Finally, 908 samples were obtained.
Results
Comment 1: Line 141-142
The soil pH values of 908 sampling points in the study area were 7.00, 8.80, and 7.85, respectively.
This sentence is not clear. Respectively is put for what? Authors want to talk about the range and mean I guest suggestion: The soil pH values of 908 sampling points in the study area were ranged from 7.00 to 8.80, with average value of 7.85.
Response 1: Thank you for your valuable suggestion. We did not describe it accurately. According to your suggestions, we have modified the error as follow (Line 167-168, Page 4):
The soil pH values of 908 sampling points in the study area were ranged from 7.00 to 8.80, with average value of 7.85.
Comment 2: How the authors explain the fact that ME of 50 samples is 0 while 75 and 100 have ME higher than 0 Line 193-194
... The closer the ME was to 0, the higher the accuracy of spatial prediction is. It can be seen from Figure 2 that the ME of 150, 200 and 300 sampling points was greater than 0 ...
From the figure 2, it is rather the 75, 100, 150 and 200 sampling points that are higher than 0; 300 being a little close to 0.
The approach is to progressively eliminate with the chosen indicators the sample points whose prediction is not very accurate compared to the 900 sampling points. This should be clearly stated in the text.
Response 2:
Thank you for your valuable suggestion. In this study, the ME value varies with the number of sampling points, but the variation does not follow the law that the ME decreases with the increase of sample number. In theory, the closer the ME was to 0, the higher the accuracy of spatial prediction. And in this case, the value of ME is generally calculated from the statistical methods. It is possible that the independent verification results of each sample point are poor, but the residual errors after plused and averaged were smaller. At this moment, the ME was close to 0, while the prediction has low accuracy. So, the index ME sometimes can't indicate the accuracy very well at this point. On the other hand, a single index can't indicate the interpolation accuracy well, and multiple indexes may be more accurate to judge the interpolation accuracy.
We have modified it in the section 3.3.
The above ME calculation formula, in which: N is the number of known samples, the actual value is Z (Xi), and the estimated value is Z'(Xi).
Comment 3: Line 216 ... 107 sampling points per 1 000 km2could meet ... put a space between km²and could
Response 3: According to your suggestion, we have put a space between km2 and could (Line 260, Page 6).
Comment 4: Figure 3: I suggest authors to put a limit on the map that should represent
I suggest that the authors put the outline of the reserve area on the map to facilitate visualization
Response 4: Thank you for your valuable suggestion. In order to show the integrity of the administrative area, the whole boundary was preserved in the spatial prediction expression. To show the approximate extent of the reserve area, we marked the unsampled area in Figure 1., which the unsampled area was almost same to the reserve area. If necessary, we will further modify our drawings to meet the requirements of the manuscript.
Discussion
Comment 1: Line 250-251 ...The ratio of Nugget to Sill of the whole sample set (908 sampling points) in this study was less than 25% ...
This statement is not true. There are 4 points whose autocorrelation (C0/Sill) was greater than 25. Be careful in wanting to generalize this result. It would be better to qualify it by saying "almost all or most of " ...
Response 1: Thank you for your valuable suggestion. According to your suggestions, we have modified the statement as follow (Line 308-309, Page 8):
The ratio of Nugget to Sill of the almost all sample set in this study was less than 25% and showed strong spatial autocorrelation ...
Comment 2: Line 279-280 ... The only literature (Wang et al., 2011) showed that the rational sampling number at the county scale was about 4 900 samples per 1 000km2 which was inconsistent with our study. Instead of study, I suggest author to use "findings"
Response 2: Thank you for your suggestion. We have revised "study" to "findings" (Line 341-342, Page 8).
Conclusion
Comment 1: Line 296-297 ... the determination coefficient of different sampling sets was between 0.39 and 0.69 ...
The determination coefficient ranged from 0.39 to 0.72. The maximum coefficient was not 0.69 the conclusion is long with description of results and/or discussion. Example lines 295 to 297; 304 to 307. This could be moved to the discussion
Response 1: According to your suggestion, we have moved the sentences in conclusion to the discussion section in lines 363-372.
References
Comment 1: One reference is missing in the references section: Wang et al., 2011
Response 1: We have added a reference as follow (Line 680-681, Page 11):
35.Wang, J.G.; Zhou, W.J.; Wang, B.W.; Chen, C. Soil Sample Density and Interpolation Accuracy on County Scale - A Case Study on Soil pH. Hunan Agricultural Sciences. 2011, 21, 27-30. [CrossRef]
In addition, we have modified (Wang et al., 2011) in the discussion section into [35]. (Line 341, Page 8)

Reviewer 2 Report
The purpose of this paper was to study the influence of diferente sampling numbers in the prediction accuracy of spatial distribution for soil pH . The results showed that the number of sampling points need to be no less than 150 sampling points. The pH value in Kenli County was generally between 7.8 and 8.1, and the soil was alkaline.
The paper is interesting and will be a good contribution for others studies.
Author Response
Responses to the comments of Reviewer #2
Dear Madam/Sir,
Thank you very much for taking the time to review our manuscript.We are very grateful for your approval of our work. I hope our work will be better and more comprehensive in the future!
Thank you again for your comments to our manuscript.
Yours sincerely,
Xiaoguang Zhang

Reviewer 3 Report
The address "Rational sampling numbers of soil pH for spatial variation: A case study from Yellow River Delta" is promising.
The topic is very attractive not only for experts, but also for a wider readership in the field of Agriculture, Civil Engineering and Environment.
The abstract is clearly written. Spatial variation in soil pH is important in assessing environmental quality. The results of this study are applicable to areas with similar environmental and soil conditions such as the Yellow River Delta. Based on the grid point distribution method, the semi-variance structure analysis was performed for different sets of points, and ordinary Kriging was used for spatial prediction and accuracy verification, and the influence of different sampling points on the spatial variation of soil pH was discussed. The results showed that the pH value in Kenli district was generally between 7.8 and 8.1 and the soil was alkaline. Semi-variance models fitted by different sets could well reflect the spatial structure characteristics of soil pH. As the number of sampling points decreased, the Sill value of the sample set increased, and the spatial autocorrelation gradually weakened. Considering the prediction accuracy, spatial distribution and investigation cost, the number of sampling points that was greater than or equal to 150 could satisfy the spatial variation expression of county-level soil pH in the Yellow River Delta. This corresponded to a number of at least 107 sampling points per 1000 km2 - OK.
Keywords
I suggest writing down only semantic keywords to represent the research, e.g., Sampling; Soil pH; Spatial variation; Ordinary Kriging; etc.
1. Introduction
In the introduction, the situation regarding the topic at hand is described. Although many scientists have conducted relevant research on the appropriate number of samplings, there are currently few reports on the related research on soil pH. Kenli County in the Yellow River Delta was selected as the study area, and the spatial distribution of soil pH with 12 different sampling sets was predicted using geostatistical Kriging interpolation. The overall objectives of study were
1) to clarify the influence of different sample numbers on the prediction accuracy of the spatial distribution of soil pH, and
2) to determine an appropriate sample density to capture the spatial variation of soil pH - OK
The references must be numbered consecutively. For example, in the introduction, reference number 9 is followed by reference number 20.
2. Materials and Methods
2.1. Study area is well described
2.2 Collection and processing of sample data
Based on the grid distribution point method, more than 1100 sampling points were distributed in this study, after reduction after collecting the samples, finally, 908 samples were obtained. Each sampling point was extracted from the last sampling point set to compare the characteristics of different sample sets. Combining all the sample data, a total of 12 sample sets were formed. The pH of each sample was measured in 1:2.5 mixtures of soil and deionized water with a pH meter – OK.
2.3 Spatial prediction and verification method
The text in 2. is partly repeated – to edit.
3. Results
3.1. Descriptive statistics characteristics of soil pH for different sets of sample points
Descriptive statistical analysis was performed on 12 sets of samples, and the results are shown in Table 1. The same values of soil pH of 908 sampling points in the study area are repeated 7.00 and 8.80 in almost all sets - is this real for measured data?
3.2 The influence of different sampling points on the semi-variance structure of soil pH
Semi-variance analysis of soil pH was carried out by geostatistical method. Table 2 shows the semi-variance function values of soil pH under different sampling points - OK.
3.3 The influence of different sampling points on the spatial prediction accuracy of soil pH - OK
3.4 Effects of different sampling points on spatial distribution of soil pH – OK
4. Discussion and 5. Conclusion
Based on discussion and conclusion an appropriate sample number of at least 150 in the Yellow River Delta region is recommended, corresponding to at least 107 sample points per 1000 km2, considering the prediction accuracy, spatial distribution, and research resources. That is correct. - OK
The results of this study are applicable to areas with similar environmental and soil conditions in the Yellow River Delta and have certain reference significance for them. However, the number of sampling points will be different to reasonably express the spatial distribution of soil pH in other areas, which needs to be analysed in conjunction with local environmental conditions - OK.
The other statements in sections 4 Discussion and 5 Conclusion seem realistic but are difficult to deduce from section 3. The discussion should be clearer (where the conclusions from section 3 come from).
The work is written in a high quality and interesting way, so I can recommend it as a scientific paper.
I suggest the authors to minor change (rewrite) the paper in light of the above comments.
I also suggest having a proof-reader check the written English.
Author Response
Responses to the comments of Reviewer #3
Dear Reviewer,
Thank you very much for your important comments, which will contribute much to the improvement of the manuscript.
We read the comments carefully and made positive revisions in the revised version. And we hope these changes would meet the requirements.
Yours sincerely,
Xiaoguang Zhang
Our replies to your comments are here as below:
Keywords
Comment 1: I suggest writing down only semantic keywords to represent the research, e.g., Sampling; Soil pH; Spatial variation; Ordinary Kriging; etc.
Response 1: Thank you for your suggestion. We have changed the keywords as follow:
Sampling; Soil pH; Spatial variation; Ordinary Kriging
1.Introduction
Comment 1: The references must be numbered consecutively. For example, in the introduction, reference number 9 is followed by reference number 20.
Response 1: According to your suggestion, we have changed the order of reference numbers (Line 63-84, Page 2).
2.Materials and Methods
Comment 1: The text in 2. is partly repeated – to edit.
Response 1: We have deleted the sentence "consisting of 908, 797, 700, 594, 499, 398, 299, 200, 149, 100, 75 and 50 samples " (Line 156-157, Page 3) and "Two verification indexes, root mean square error and average error" (Line 159-160, Page 4).
3.Results
Comment 1: Descriptive statistical analysis was performed on 12 sets of samples, and the results are shown in Table 1. The same values of soil pH of 908 sampling points in the study area are repeated 7.00 and 8.80 in almost all sets - is this real for measured data?
Response 1: This is real for measured data. Based on the grid distribution point method, 1 140 sampling points were set in this research. These sampling points were mainly distributed in cultivated land, and the interval between sample points was about 600 m. In the actual sampling process, we have adjusted the specific positions of these sampling points for road accessibility and crop planting. Although all these samples were sampled, there were many uncertainties and complexity in field sampling. Consequently, we eliminated some points that were not standardized in the collection process, and removed some outliers. Finally, 908 samples were obtained. Based on 908 sample points, 797 samples were randomly extracted to form a new sample set. Then, from 797 sample points, 700 sample points were randomly selected to form a new sample set. By parity of reasoning, each sampling is randomly selected from the previous sample set. In this way, each sampled point set will have the same characteristics as the original different sampling point sets.
We also revised it in 2.2.
4. Discussion and 5. Conclusion
Comment 1: The other statements in sections 4 Discussion and 5 Conclusion seem realistic but are difficult to deduce from section 3. The discussion should be clearer (where the conclusions from section 3 come from).
Response 1: We have made changes to make the discussion and conclusion clearer. Lines 363-372 have been modified.
Comment 2: I also suggest having a proof-reader check the written English.
Response 2: Considering the Reviewer's suggestion, we have made great efforts to modify the sentence to make it more professional. Our manuscript has been revised for proper English language, grammar, punctuation, spelling, and overall style by Pro. Dr Xiangyun Song who was abroad for many years. And I hope it can meet the requirements.
